# HDL Proteome and Alzheimer’s Disease: Evidence of a Link

**DOI:** 10.3390/antiox9121224

**Published:** 2020-12-03

**Authors:** Judit Marsillach, Maria Pia Adorni, Francesca Zimetti, Bianca Papotti, Giovanni Zuliani, Carlo Cervellati

**Affiliations:** 1Department of Environmental & Occupational Health Sciences, University of Washington, Seattle, WA 98195, USA; jmarsi@uw.edu; 2Unit of Neurosciences, Department of Medicine and Surgery, University of Parma, 43126 Parma, Italy; mariapia.adorni@unipr.it; 3Department of Food and Drug, University of Parma, 43124 Parma, Italy; bianca.papotti@unipr.it; 4Department of Morphology, Surgery and Experimental Medicine, University of Ferrara, 44121 Ferrara, Italy; giovanni.zuliani@unife.it (G.Z.); crvcrl@unife.it (C.C.)

**Keywords:** Alzheimer’s disease, inflammation, vascular dementia, high-density lipoprotein, accessory proteins, apolipoprotein A-I, apolipoprotein E, apolipoprotein J, paraoxonase 1

## Abstract

Several lines of epidemiological evidence link increased levels of high-density lipoprotein-cholesterol (HDL-C) with lower risk of Alzheimer’s disease (AD). This observed relationship might reflect the beneficial effects of HDL on the cardiovascular system, likely due to the implication of vascular dysregulation in AD development. The atheroprotective properties of this lipoprotein are mostly due to its proteome. In particular, apolipoprotein (Apo) A-I, E, and J and the antioxidant accessory protein paraoxonase 1 (PON1), are the main determinants of the biological function of HDL. Intriguingly, these HDL constituent proteins are also present in the brain, either from in situ expression, or derived from the periphery. Growing preclinical evidence suggests that these HDL proteins may prevent the aberrant changes in the brain that characterize AD pathogenesis. In the present review, we summarize and critically examine the current state of knowledge on the role of these atheroprotective HDL-associated proteins in AD pathogenesis and physiopathology.

## 1. Introduction

Dementia is a syndrome now affecting over 40 million people worldwide, and is one of the major causes of disability and death in older people [1]. In particular, Alzheimer’s disease (AD) is the most frequent cause of dementia in elderly populations, with nearly 70% of patients [1].

The key pathological changes observed in AD brain tissues are the deposition of amyloid-β (Aβ) peptides (Aβ1-40 and Aβ1-42) in diffuse and neuritic plaques, and of intracellular neurofibrillary tangles (NFT), primarily consisting of abnormal and hyper-phosphorylated tau protein [2]. Additional frequent abnormalities are represented by the presence of neuroinflammation and cerebral vessel disease, with the latter affecting 60–90% of AD patients [3].

It has become increasingly clear that the role of vascular dysregulation in AD development is beyond that of a “static” risk factor and/or common comorbidity [4]. Indeed, ischemic and neurodegenerative pathology seem to reciprocally interact, from the earliest stage of AD, affecting the clinical presentation and the progression of the disease [4]. As a logical consequence of this hypothesized link, every cardiometabolic risk factor can be a candidate clinical predictor of AD. High-density lipoprotein cholesterol (HDL-C) is an emblematic example in this regard. This inverse predictor of cardiovascular disease (CVD), which emerged from epidemiological studies, has shown a similar relationship in several prospective studies on AD, where higher levels of plasma HDL-C were associated with lower risk of developing the disease [5]. Consistently, some large cross-sectional studies have reported lower levels of HDL-C in AD patients. However, contrasting results have also been reported on this association [6,7].

In support of an association between HDL and AD, recent evidence has supported the role of HDL in preserving cognitive functions during aging. In fact, a centenarians study in Ashkenazi Jews revealed that plasma HDL-C levels are positively related with cognitive function, determined by the mini-mental state examination (MMSE), an index of cognitive performance [8]. In accordance, the Leiden 85 plus study, a population-based study in patients from The Netherlands who reached 85 years of age, indicated that median MMSE scores were significantly lower in subjects with low HDL-C, independently of CVD disease [9]. In addition, low HDL-C levels were related to a decline in memory, defined by short-term verbal memory, compared with high HDL-C levels in middle-aged adults in the Whitehall II study (London). The reported decrease in HDL-C levels was associated with memory decline independently of confounding factors, including education, employment, comorbidity, and APOE genotype, over the 5 years of follow-up [10].

A French cohort study in AD patients and control subjects evidenced lower plasmatic HDL-C levels in AD patients compared to controls [6]. Moreover, the prospective cohort, Manhattan study found that higher HDL-C levels (>55 mg/dL) were associated with a decreased risk of both probable and possible AD (hazard ratio, 0.4; 95% confidence interval, 0.2–0.9; *p* = 0.03), also after adjustment for age, sex, education, ethnic group, and *APOE4* genotype [11]. Accordingly, another study in centenarian Ashkenazi Jews (mean age of 99 years), carriers of the polymorphism 1405 V on the *cholesteryl ester transfer protein* (*CETP*) gene, demonstrated that low CETP levels, increased levels of HDL-C, and large HDL particles were significantly correlated with peculiar longevity, with a MMSE score in the normal range (>25 points) [12].

Nevertheless, genetic polymorphisms and pharmacological strategies influencing HDL-C levels have not consistently associated with CVD risk [13,14,15,16]. Similarly, Mendelian randomization studies also suggested that HDL-C levels are not a causal risk of AD [17]. Notably, these investigations addressed only a causal relationship between disease risk and increased HDL-C levels mediated by selected genes. As elegantly highlighted by Button et al. [5], all of these studies did not examine the changes in HDL function that can occur in CVD or AD onset and progression and that can be much superior predictors of disease risk [5]. Indeed, the functional properties of HDL, and in particular the capacity of HDL to promote cholesterol efflux from macrophages, the first step of the reverse cholesterol transport, is emerging as a better predictor of CVD and in the majority of the cases independently of plasma levels of HDL-C [18,19].

HDLs are a highly heterogeneous lipoprotein family, consisting of several subclasses differing in size, shape, and lipid and protein composition [13,20]. The biological functions of HDLs are not limited to their role in cholesterol metabolism, but include several other vasoprotective actions, such as the enhancement of vascular function and tone, the prevention of inflammation, the counteraction of lipoperoxidation and the induction of nitric oxide secretion release and endothelial repair [20,21]. It is widely accepted that HDL function is closely related to the composition and quality of its proteome, which includes at least 204 members [22]. Apolipoprotein A-I (apoA-I) is the most abundant protein of HDLs, and its integrity is known to be pivotal for HDL global function [23]. Other important apolipoproteins are apoA-2, apoE, and apoJ, exerting various tasks in cholesterol metabolism [23].

The oxidation of apoA-I leads to a dysfunctional HDL. Martínez-López et al. [24] has recently shown that this modification is linked to impairment in cholesterol efflux and lecithin cholesterol acyltransferase (LCAT)-activating activities of the lipoprotein. There are some other components of the HDL proteome that protect apoA-I from oxidation. Among these, Paraoxonase 1 (PON1) seems to be the most effective [25]. PON1 is one of the main contributors to the atheroprotective properties of HDLs, and decrease in its activity is associated to higher risk of developing CVD, as well as other diseases where oxidative stress (OxS) or inflammation play a pathogenic role, including AD [26,27,28]. Although the crystal structure of human PON1 is not available yet, several convergent pieces of evidence have indicated that it is physically associated to apoA-I with both proteins modulating their activity reciprocally [29,30,31].

Some intriguing hypothesis on the role of HDL and its associated proteins in brain function and their influence on neurodegeneration, have also been formulated [5,32,33]. This influence could be indirect, due the reported beneficial effects of HDLs on the cardiovascular system, then reflected in the brain [5]. More intriguingly, the crosstalk between circulating HDL and the brain can occur via diffusion of apolipoproteins from the periphery to the central nervous system (CNS). Alternatively, the apolipoproteins may also be synthetized in situ. Indeed, apoA-I and apoE are present within the brain parenchyma, in cerebrospinal fluid (CSF) and in the cerebrovascular intima layer of the leptomeningeal artery [34]. In CSF, these two apolipoproteins are constituents of lipoproteins with a size and density similar to plasma HDL (HDL-like), which transport lipids in the brain [35]. In the present review, we summarize and critically examine the current state of knowledge on the role of selected HDL-associated proteins (Figure 1) in AD pathogenesis and physiopathology, especially focusing on under investigated aspects and suggesting potential novel mechanisms of action. A deeper knowledge on how HDL-proteins influence cerebral functions may provide promising prospective methods for the identification of novel therapeutical opportunities in neurodegenerative diseases.

## 2. HDL Apolipoproteins

### 2.1. Apolipoprotein A-I

ApoA-I represents the major protein constituent of HDL (70% of HDL protein content) and it is crucial for the structural and functional integrity of the lipoprotein [36]. It is indeed largely responsible for mediating HDL assembly and is a determinant of HDL structure and composition. Moreover, this apolipoprotein plays a critical role in reverse cholesterol transport by enhancing the cholesterol efflux from macrophages, through an interaction with the ATP-binding cassette transporter, A1 (ABCA1). Finally, apoA-I is also a key stimulator of the enzyme LCAT, which is pivotally involved in the esterification of free cholesterol and HDL particle remodeling [37].

#### 2.1.1. Major Evidence Supporting a Role of Apolipoprotein A-1 in Alzheimer’s Disease Pathogenesis

The apoA-I present in the brain is thought to be primarily derived from the peripheral circulation through both the blood–cerebrospinal fluid barrier (BCSFB) and the blood–brain barrier (BBB) [34]. Several studies have suggested that apoA-I plays a critical role in preserving cerebrovascular integrity and reducing AD risk (Table 1).

#### 2.1.2. ApoA-I and Aβ

Some pre-clinical studies highlighted the role of apoA-I in AD using different transgenic mouse models of AD that were used to explore the metabolism of human lipoproteins. For example, APP/PS1 transgenic mice overexpressing human apoA-I were characterized by an increase in plasma HDL-C levels, and a parallel improvement of memory deficit and attenuation of Aβ-associated neuroinflammation and cerebral amyloid angiopathy [32]. In accordance, Lefterov et al. [38] demonstrated that the deletion of apoA-I in the same mouse model worsened memory impairment. Moreover, in agreement with Lefterov et al., in a recent study by Button and colleagues [39], it was observed that complete loss of apoA-I increases cerebral amyloid angiopathy in APP/PS1 mice.

It has also been reported that apoA-I can affect the amyloid precursor protein (APP) processing pathway. In particular, the hypothesis is that apoA-I and HDL mediate the cholesterol efflux process, resulting in enhanced cell membrane fluidity that increases the non-amyloidogenic cleavage by α-secretase to generate the soluble form of Aβ, which undergoes hepatic clearance and does not accumulate in plaques [40]. Alternatively, apoA-I could bind APP on the cell surface, avoiding the APP endocytic process, essential for β- and γ-secretases activities and resulting in a reduced production of the neurotoxic, insoluble Aβ. The interaction between Aβ and apoA-I with respect to AD pathogenesis has been recently extensively reviewed by Ciccone L. et al. [33]. A mechanism implicated in the neuroprotective effect of HDL could be the enhancement of Aβ clearance. In this regard, in vitro evidence suggests that apoA-I interacts with Aβ and prevents its aggregation [40]. In addition, apoA-I and HDL have a high affinity for Aβ and mediate its clearance through the BBB [41]. In this respect, it has been recently reported that apoA-I lipidation state, and its conformation, could affect Aβ clearance from the brain. In particular, discoidal HDL had a higher capacity to promote Aβ efflux across the BBB in vitro than apoA-I in different lipidation states. In addition, discoidal HDL may disturb Aβ conformation by decreasing fibril concentration and extension [42]. Recently, Contu L. et al. [43] reported that in the Tg2576 mouse model, deletion of apoA-I is associated with increased clearance of Aβ.

In human epidemiological studies, a French cohort study highlighted lower plasma apoA-I levels in AD patients compared with controls [6]. In the study from Slot et al. [44], higher CSF and lower plasma apoA-I levels were associated with an increased risk of clinical progression in subjects with mild cognitive impairment (MCI). In contrast, in another study [45], CSF levels of apoA-I were reduced in AD patients compared to controls. Finally, the phase II clinical trial, ASSERT (a stratified sickle event randomized trial), in which patients were treated for 12 weeks with RVX-208, a small molecule that stimulates APOA-I gene expression, showed an increase in plasma Aβ levels compared to baseline. Thus, the increase in plasma apoA-I levels was associated with enhanced Aβ clearance from the brain [46], with a peripheral “sink effect” that resulted in decreased Aβ burden [47].

**Table 1 antioxidants-09-01224-t001:** Selection of preclinical and clinical evidence supporting the role of apolipoprotein A-I (apoA-I) in protecting from Alzheimer’s disease (AD).

Study	Type of Study	Main Findings
Preclinical Studies		
Koldamova, R.P. et al., 2001 [40]	In vitro	ApoA-I directly interacted with amyloid precursor protein and inhibited Aβ aggregation and toxicity
Dal Magro, R. et al., 2019 [42]	In vitro	ApoA-I mediated clearance of Aβ through the blood-brain barrier
Contu, L. et al., 2019 [43]	In vivo (animal model)	Deletion of apoA-I was associated with increased clearance of Aβ and reduced parenchymal and vascular Aβ pathology
Paternò, L. 2004 [48]	In vivo (animal model)	Infusion of reconstituted HDL containing apoA-I ameliorated the neuronal impairment through an antioxidant mechanism
Lewis, T.L. et al., 2010 [32]	In vivo (animal model)	APP/PS1 transgenic mice overexpressing human apoA-I showed an improvement of memory deficit and reduction of Aβ deposition
Lefterov, I. et al., 2010 [38]	In vivo (animal model)	Deletion of apoA-I in APP/PS1 transgenic mice worsened memory impairment
Button, E.B. ret al., 2019 [39]	In vivo (animal model)	Loss of apoA-I in APP/PS1 mice increased total cortical Aβ deposition and astrogliosis
Clinical/Epidemiological Studies		
Merched, A et al., 2000 [6]	Cross-sectional	Lower apoA-I levels in Alzheimer’s disease AD patients compared to controls; apoA-I levels correlated with disease severity (*n* = 157)
Slot, R.E.R. et al., 2017 [44]	Longitudinal	Higher cerebrospinal fluid (CSF) levels of apoA-I were associated with an increased risk of clinical progression in non-demented individuals (*n* = 429)
Johansson, P. et al., 2017 [45]	Cross-sectional	Lower CSF levels of apoA-I in AD patients compared with controls; apoA-I levels correlated with the mini-mental state examination MMSE score and phospho-tau (p-tau) (*n* = 74)

#### 2.1.3. ApoA-I, Neuroinflammation, and Oxidative Stress

Beyond the direct action on Aβ, further mechanisms that contribute to the protective role of HDL could be related to the antioxidant and anti-inflammatory properties of HDL. In this context, it has been observed that reconstituted HDL containing apoA-I, if administered in a rat model of stroke, ameliorated the neuronal impairment through an antioxidant mechanism by reducing reactive oxygen species (ROS) [48]. Moreover, a triple transgenic mouse model overexpressing APP and PS1 mutations together with human apoA-I (APP/PS1/AI) showed that the presence of apoA-I counteracted learning and memory impairment and reduced neuroinflammation. In particular, APP/PS1/AI mice displayed increased HDL-associated paraoxonase activity [32] and a decreased microglia–astrocyte activation compared to APP/PS1 mice. In accordance, cultured Aβ-induced hippocampal slides overexpressing apoA-I also showed a decreased production of the inflammatory cytokines monocyte chemoattractant protein-1 (MCP-1) and interleukin-6 (IL-6) [32]. Additionally, Button et al. [39], have recently shown that a loss of apoA-I in APP/PS1 mice increased several markers of neuroinflammation and vascular inflammation within the brain. The antioxidant activity of apoA-I could be related to the reported chaperone activity of the apoprotein, that, by complexing with Aβ, prevents its pro-oxidant effect [49]. In addition, apo A-I may potentiate the antioxidant activity of PON1, as discussed below. Interestingly, among the potential mechanisms explaining the anti-neuroinflammatory potential of apoA-I, there could be a competitive activity with the internalization of the pro-inflammatory protein serum, amyloid A (SAA), mediated by the scavenger receptor class B type I (SR-BI). In support of this hypothesis are the expression of SR-BI in human astrocytes [50] and the observation that both apoA-I and SAA are SR-BI ligands [51,52].

### 2.2. Apolipoprotein E

Apolipoprotein E (apoE) is a glycoprotein mainly synthesized by the liver, but also by several peripheral tissues, including the brain. The *APOE* gene displays a genetic polymorphism determined by three common alleles: *APOE2*, *APOE3*, and *APOE4* [53]. The three isoforms differ only by a single amino acid substitution, leading to altered protein structure and correlated biological functions. Specifically, the *APOE4* allele is widely known to be one of the strongest risk factors for developing AD [54], while *APOE2* has shown neuroprotective effects [55].

ApoE is one of the major components of the HDL-like particles in the CNS, even though being present at much lower concentration compared to plasma HDL. The lipoprotein particles containing apoE are usually unable to cross the BBB, so peripheral and CNS apoE can be considered as two distinct pools [56].

#### 2.2.1. Major Evidence Supporting a Role of Apolipoprotein E in Alzheimer’s Disease Pathogenesis

The involvement of apoE in AD pathogenesis occurs through multiple mechanisms, including an influence on Aβ metabolism, but also on the tau protein and the synaptic function, as well as on neurotoxicity. In addition, apoE possesses several additional activities, including the capacity to influence brain lipid metabolism, neuroinflammation, and oxidative stress. Many of these properties, that differ among the apoE isoforms, contribute to the protective functions of the brain apoE and apoE-containing particles, with respect to neurodegenerative diseases, similarly to what occurs for plasma HDL in CVD [57]. The most important evidence linking apoE and AD is reported in Table 2.

#### 2.2.2. ApoE and Aβ

The role of apoE in Aβ processing and deposition is very well known, and has been extensively reviewed elsewhere [58]. Specifically, apoE binds Aβ, forming complexes able to modify Aβ clearance, aggregation and thus the formation of the plaques. Within this context, the isoform apoE4 showed increased Aβ binding ability, and increased Aβ oligomerization rate [59]. In addition, apoE4 isoform may compete with Aβ for interaction with the LDL receptor-related protein 1 (LRP1), leading to reduced Aβ clearance and increased brain deposition [60]. This mechanism would explain the observations made in postmortem human AD brains, where the positive correlation between LRP1 and Aβ levels was strengthened by the presence of the *APOE4* allele [61].

#### 2.2.3. ApoE and Tau

ApoE has also an influence on tau and tau-mediated neurodegeneration [58]. The isoform apoE4 worsened tau-mediated neurodegeneration in mouse models [62] and correlated with higher tau aggregates in the brain of post-mortem individuals [63]. Mechanistically, apoE4 may induce phosphorylated tau and cell injury through a heparin sulfate proteoglycan-dependent process [64].

#### 2.2.4. ApoE and Fragmentation

Another important mechanism explaining the involvement of apoE in AD relates to the proteolytic cleavage that the protein undergoes in the brain, generating truncated fragments [65]. In particular, the isoform apoE4 is subjected to an enhanced proteolysis compared to apoE3, generating neurotoxic fragments [66]. In humans, apoE fragments have been detected in the AD brain at higher concentrations compared to non-AD controls, following an *APOE* gene-dose-dependent pattern [67].

#### 2.2.5. ApoE and Cholesterol Homeostasis

One of the main roles of apoE in the CNS is the maintenance of cholesterol homeostasis in the brain, which occurs through the transport of newly synthetized cholesterol from astrocytes to neurons [58]. In fact, adult neurons lose endogenous synthesis capacity, and rely on cholesterol provided by other cells to ensure physiological functions [68]. Brain cholesterol transport from astrocytes to neurons occurs with astrocyte-secreted apoE that binds cholesterol and phospholipids through an interaction with the membrane transporters ABCA1, ABCG1, and remodeling enzymes [35], generating particles able to interact with the neuronal LDL receptors family, and leading to cholesterol internalization [69]. Proper apoE lipidation is thus important to preserve the brain HDL-like particle function, and the lipidation degree of these particles appears to be isoform-dependent, with apoE4 being poorly lipidated compared with apoE2 and apoE3 [70]. This results in the formation of smaller particles, as has been detected in the CSF of *APOEε4* carriers when compared with *APOE3* individuals [70]. Lipidation of apoE is mediated by ABCA1 activity and, in this regard, cholesterol efflux from apoE4- and ABCA1-expressing astrocytes was lower compared to the efflux from apoE3-expressing cells [71]. A possible mechanistic explanation for this difference has been provided by Rawat and colleagues; they found that apoE4 displayed a reduced interaction with ABCA1 and therefore a lower lipidation degree compared with apoE3. Specifically, apoE4 induced trapping of ABCA1 intracellularly, leading to a lower ABCA1-mediated cholesterol efflux and lower Aβ degradation capacity [72]. In humans, we and others [73,74] have demonstrated that the CSF from mild cognitive impairment (MCI), and AD, patients showed a lower ability to induce ABCA1- and ABCG1-efflux.

As mentioned above, the HDL-like particles interact with the LDLr and other APOE-binding receptors, promoting cholesterol uptake by neurons. Alterations in the interaction between apoE and its receptors may have potential deleterious impacts on neuronal functions. For example, the proprotein convertase subtilisin/kexin type 9 (PCSK9), very well known for its regulating effect on plasma lipids [75], seems to promote the degradation of the apoE-receptors in the brain [76]. In this regard, we and others found elevated PCSK9 concentrations in the CSF of AD patients [77]. In addition, an increased PCSK9 expression in the frontal cortex from AD patients, and an association between the presence of a specific PCSK9 SNP and the risk of AD, were observed [78].

Interestingly, in individuals with a parental history of AD, the CSF apoE levels correlate with those of PCSK9 [78], strengthening the hypothesis that PCSK9 induces the degradation of the brain apoE-receptors, leading to increased CSF apoE concentrations. In addition, a peculiar link between apoE4 and PCSK9 emerged in our study, in which CSF PCSK9 was higher in the apoE4 isoform carriers compared to non-carriers [77]. In line with this observation, individuals bearing the apoE4 isoform showed a breakdown of the BBB [79]. Based on these data we hypothesized that increased BBB permeability may facilitate the crossing of PCSK9 from the periphery, possibly helping to explain the higher concentration in the brain of apoE4 carriers. Supporting this hypothesis, the administration of a small molecule, inhibiting PCSK9, restored BBB proteins in the hippocampus, and decreased cognitive impairment in rats [80].

**Table 2 antioxidants-09-01224-t002:** Selection of preclinical and clinical evidence supporting the role of apolipoprotein E (apoE) in protecting from Alzheimer’s disease (AD).

Study	Type of Study	Main Findings
Preclinical Studies		
Verghese, E.P. et al., 2013 [60]	In vitro	ApoE4 may compete with Aβ for the interaction with the LDL receptor-related protein 1 (LRP1), leading to reduced Aβ clearance, increasing brain Aβ deposition
Pocivasek, A. et al., 2009 [81]	In vitro	In LPS-treated glial cells, apoE displayed an anti-inflammatory action
Wong, M.Y. 2020 [82]	In vitro	Increased secretion of IL-1β secretion in apoE4- compared with apoE3-expressing microglia
Shi, Y. et al., 2017 [62]	In vivo (animal model)	ApoE4 worsened the tau-mediated neurodegeneration
Lynch, J.R. et al., 2003 [83]	In vivo (animal model)	APOE4 expression associated to higher systemic and brain elevations of the TNF-α and IL-6.
Farfel, J.M. et al., 2001 [63]	Post-mortem (humans)	ApoE4 was associated with higher tau aggregates in the brain
Mouchard, A. et al., 2019 [67]	Post-mortem (humans)	ApoE neurotoxic fragments were increased in the brain of AD compared with controls: these fragments form heteromers with Aβ, slowing down its clearance
Ramassamy, C. et al., 2000 [84]	Post-mortem (humans)	Higher markers of oxidative stress in the hippocampus of APOE4 AD patients
Clinical/Epidemiological Studies		
Rawat, V. et al., 2000 [72]	Cross-sectional	APOE4 carriers showed a reduced CSF ABCA1-cholesterol efflux (*n* = 20)
Marchi, C. et. al, 2019 [73] Yassine H.N. et al., 2016 [74]	Cross-sectional	CSF from mild cognitive impairment and AD patients showed a lower ability to induce ABCA1- and ABCG1-mediated cholesterol efflux (*n* = 200)
Tao, Q. et al., 2018 [85]	Longitudinal	Only in apoE4 carriers, the presence of chronic low-grade inflammation was associated with increased risk of AD (*n* = 2656)
Di Domenico, F. et al., 2016 [86]	Cross-sectional	Presence of oxidative products in the CSF of mild cognitive impairment (MCI) and AD patients compared to controls (*n* = 18)

#### 2.2.6. ApoE and Neuroinflammation

An anti-inflammatory role for apoE has been documented by several studies, and recently reviewed [59]. Among them, in lipopolysaccharide (LPS)-treated glial cells, apoE displayed an anti-inflammatory action, occurring through the c-Jun N-terminal kinase (JNK)-mediated pathway [81].

Other mechanisms, that are still to be investigated, may include a direct microglial effect on the transcription factor Nf/kB or the inflammasome, as occurs for peripheral HDL on macrophages [87]. The anti-inflammatory effect of apoE was also highlighted in vivo, with apoE4 being associated with the highest increase of tumor necrosis factor-alpha (TNF-α) and IL-6 in mouse brains [83]. In humans, the presence of chronic low-grade inflammation was associated with increased risk of AD, and to shorter latency of disease onset, only in APOE4 carriers [85].

#### 2.2.7. ApoE and Oxidative Stress

ApoE may also influence oxidative stress in an isoform-dependent manner. This relationship has been demonstrated in the majority of the studies conducted in *APOE* knockout (KO) mice, and reviewed by Butterfield and colleagues [88]. In the hippocampus of AD patients, markers of oxidative stress were higher in *APOE4* carriers [84]. In addition, a proteomic analysis of CSF from MCI subjects revealed the presence of oxidation products before dementia was diagnosed [86].

Moreover, in human apoE4 targeted replacement mice, a reduction in the levels of the endogenous antioxidant thioredoxin-1 was observed [89], suggesting a direct effect of apoE4 on this enzyme. Other mechanisms underpinning the pro-oxidant activity of apoE4 may be related to a weaker interaction with Aβ, impeding the mitigation of its oxidative power, or a reduction of the activity of antioxidant enzymes. All these hypotheses need to be confirmed by future studies.

### 2.3. Apolipoprotein J

Apolipoprotein J (apoJ), also known as clusterin, is a human 80 kDa glycoprotein first isolated in testis fluid, and identified for its aggregating or “clustering” effect on Sertoli cells [90]. In the following years, clusterin was found in different tissues under different names, according to tissue localization and function including: testosterone repressed prostate messenger-2 (TRPM-2), serum protein-40,40 (SP-40,40) [91], complement cytolysis inhibitor (CLI), sulfated glycoprotein 2 (SGP-2), and apoJ [92]. Finally, in 1992 it was concluded that all these proteins were produced from the same gene, called *CLU* [93]. ApoJ is expressed in different tissues, such as the pancreas, lymphoid tissue, testis, prostate, and brain [92], and is present in biological fluids; apoJ is associated with a variety of functions including complement inhibition [91], chaperon function [94], lipid transport [95], and apoptosis [96].

#### 2.3.1. Major Evidence Supporting a Role of Apolipoprotein J in Alzheimer’s Disease Pathogenesis

In the CNS, apoJ is mainly produced by astrocytes [97], and it seems to contribute to the clearance of Aβ from the brain, due to the ability of ApoJ-containing lipoproteins to bind Aβ, and to be rapidly eliminated across the BBB, via low density lipoprotein-related protein 2 (LRP2) receptor [47]. The most important evidence linking apoJ and AD is reported in Table 3.

Two genomic wide association studies (GWAS) have identified the *CLU* gene as a novel risk factor for late-onset AD [98,99], becoming the third most common genetic risk factor after *APOE* and *bridging integrator 1* (*BIN1*). Subsequently, genetic studies have discovered several single nucleotide polymorphisms (SNPs) as susceptible loci [100]. Recently, a common *CLU* variant (rs9331896) has been genotyped and associated with a high risk of AD and all dementia in the general population [101].

The association between apoJ concentration in brain tissue, CSF, serum, and plasma, and dementia has been extensively reviewed in a recent systematic review and meta-analysis [102]. It was shown that apoJ concentration in the plasma and brain tissues was increased in dementia compared to control, and that plasma levels of apoJ were significantly increased only in the AD group, and not in MCI or other dementias. Moreover, no association was found between serum and CSF apoJ concentration and dementia. In this meta-analysis it was thus concluded that high apoJ concentration, both in the plasma and brain, is associated with dementia. In line with this evidence, significant increases in both intracellular and secreted apoJ were found in the brain tissue of individuals with AD, and the increase in apoJ alloforms was significantly associated with increases in both insoluble Aβ42 and tau protein [103]. Furthermore, apoJ concentrations were also higher in the hippocampus and cortex of the AD brain together with Aβ plaques [104], and this also occurred in the CSF [105]. Wang and colleagues [106] found that CSF apoJ levels were positively correlated with neurogranin (NG), an index of synaptic degeneration, in the control and MCI groups, but not in the AD group. However, in all subjects, apoJ levels were positively associated with NG levels, independently of age, gender, *APOE4* genotype, clinical diagnosis, and CSF Aβ42 levels. In addition, increased plasma apoJ levels have been correlated with cortex atrophy, and were also associated with a greater burden on fibrillar Aβ in the brain [107].

**Table 3 antioxidants-09-01224-t003:** Selection of preclinical and clinical evidence supporting the role of apolipoprotein J (apoJ) in protecting from Alzheimer’s disease (AD).

Study	Study Type	Main Findings
Preclinical Studies		
Yerbury, J.J. et al., 2010 [108]	In vitro	Treatment with apoJ preserved cell viability of neuroblastoma cells treated with CSF supplemented with Aβ
Nielsen, H.M. et al., 2010 [109]	In vitro	Treatment with apoJ reduced Aβ oligomer uptake in human astrocytes
Bell, R.D. et al., 2007 [47]	In vivo (animal model)	ApoJ contributed to the clearance of Aβ from the brain
May, P.C. et al., 1990 [104]	In vivo (animal model)	ApoJ concentrations were increased in the AD brain, together with Aβ plaques
Cascella, R. et al., 2013 [110]	In vivo (animal model)	Treatment with apoJ before the injection of Aβ aggregates in the brain improved learning and memory performance
DeMattos, R.B. et al., 2002 [111]	In vivo (animal model)	Deletion of *CLU* gene led to reduced amyloid deposition and neuritic dystrophy
Qi, X.M. et al., 2018 [112]	In vivo (animal model)	Administration of an apoJ mimetic peptide led to an improvement of cognitive function and a reduction in Aβ plaque deposition
Shepherd, C.E et al., 2020 [103]	Post-mortem (human)	Both intracellular and secreted apoJ were increased in the AD brain; the increase in apoJ alloforms was positively associated with Aβ42 and tau levels
Clinical/Epidemiological Studies		
Tan L. et al., 2016 [100]	Genetic	*CLU* genotypes modulated the cerebral Aβ loads in hippocampus (*n* = 812)
Nordestgaard, L.T. et al., 2018 [101]	Genetic	*CLU* variant was associated with a high risk of AD and all dementia (*n* = 362, 338)
Yang, C. et al., 2019 [102]	Meta-analysis (genetic studies)	High apoJ concentration in the plasma and brain was associated with dementia, especially in AD patients (*n* = 28 studies)
Wang, J. et al., 2020 [106]	Cross-sectional	CSF apoJ levels were positively correlated with markers of synaptic degeneration in non-demented subjects and patients with MCI (*n* = 294)
Thambisetty, M. et al., 2010 [107]	Cross-sectional	High plasma apoJ levels were positively associated with cortex atrophy (*n* = 844)

#### 2.3.2. ApoJ and Aβ

With respect to the effect of apoJ on amyloid aggregation, and the consequent formation of amyloid fibrils and neuronal toxicity, a study revealed the ability of apoJ to bind different oligomers of Aβ, and to create long-term stable complexes, thereby influencing Aβ aggregation and the formation of Aβ fibrils [113]. Therefore, the ability of apoJ to sequester toxic oligomers may provide a molecular basis for the recently identified genetic association between apoJ and AD [98,99]. Indeed, conditions of reduced apoJ levels or reduced apoJ ability to form stable complexes with Aβ oligomers, are potentially associated to an increased susceptibility of the individual to develop AD. Consistently, the addition of a mix of physiological concentrations of a few chaperones, including apoJ, preserved cell viability of neuroblastoma treated with CSF supplemented with Aβ, and enhanced Aβ uptake by macrophage-like cells [108]. In the same way, a study on rat primary hippocampal cells demonstrated that pretreatment with apoJ prevented the increase of intracellular calcium, ROS generation, and proapoptotic caspase-3 trigger, by forming complexes with Aβ [114]. In another study, the treatment with apoJ before the injection of Aβ aggregates in the rat brain hippocampus, avoided Aβ-induced injury, and improving learning and memory performance, measured with the Morris maze test, and decreased inflammation and neuronal degeneration in rat brains [110].

On the other hand, with respect to the effect of apoJ on Aβ clearance, numerous mechanisms have been identified; among these the promotion of Aβ intracellular uptake and transport across the BBB have been reported. On the other hand, treatment with apoJ reduced Aβ oligomer uptake in human astrocytes incubated with Aβ in vitro [109]. Regarding the implication of apoJ in Aβ transport across the BBB, Aβ clearance was enhanced when aggregated with apoJ in vivo, through a mechanism mediated by the LRP2 receptor [47]. Recent findings obtained using an in vitro BBB model of primary cerebral endothelial cells, cultured on trans-wells to mimic the trafficking between the basolateral [115] and apical (blood) compartments, showed that the transport of labelled fluorescent Aβ from the basolateral to apical compartment was increased when complexes with apoJ were formed. This passage decreased when the LRP1/LRP2 pathway was blocked [116], suggesting that the relationship between apoJ and Aβ might be relevant for amyloid clearance in vivo.

By using animal models of amyloidosis, the correlation between apoJ and Aβ in vivo has been investigated, however conflicting results have been achieved. Unexpectedly, a study by DeMattos et al. [117] showed that PDAPP mice, a transgenic mouse model of AD, which were bred to *APOJ* KO mice, displayed significantly reduced fibrillar amyloid deposition compared to apoJ-expressing PDAPP mice. This phenomenon was accompanied by a decreased neuritic dystrophy, which points to a pro-amyloidogenic role of apoJ. Subsequently, a second work from the same group demonstrated that the absence of both apoJ and apoE significantly increased Aβ production and amyloid deposition [111].

Moreover, in another study the intracerebroventricular administration of an apoJ mimetic peptide in AD transgenic mice for 2 weeks, resulted in an improvement of cognitive function assessed by the water maze test. Immunohistochemistry analyses also revealed that amyloid plaque deposition was reduced in the apoJ-treated group, together with a reduction of brain soluble Aβ40 and Aβ42 levels. Finally, the treatment increased expression of LRP2, a receptor involved in Aβ clearance from the brain, in the hippocampus and temporal cortex, [112]. Consistently, peripheral administration of human recombinant apoJ/clusterin in APP23 mice, induced a reduction of insoluble Aβ and cerebral amyloid angiopathy in the brain [118]. In the same study an anti-neuroinflammatory effect of apoJ treatment was also observed.

In light of the above described data, apoJ certainly plays a role in AD pathogenesis via various processes, including aggregation and clearance of Aβ, neuroinflammation and lipid metabolism modulation, and regulation of the neuronal cell cycle and apoptosis. Interestingly, apoJ levels are increased in AD and it decreases the aggregation of Aβ. However, growing evidence shows that when Aβ levels are much higher than those of apoJ, the amyloid generation is increased. Moreover, amyloid aggregates incorporating apoJ are more toxic compared to the Aβ aggregates alone [119]. Thus, apoJ may potentially offer therapeutic opportunities to regulate Aβ load in the brain, in opposition to neurodegeneration. Nevertheless, given the numerous activities carried out by this apolipoprotein, including neuroprotective and pathogenic mechanisms, it should not to be considered as a traditional therapeutic target, and further studies need to be carried out to explore novel potential apoJ-based therapeutic strategies in AD.

## 3. HDL Accessory Proteins

### 3.1. Paraoxonase 1 (PON1)

Paraoxonase 1 (PON1) is an antioxidant, anti-inflammatory, and anti-apoptotic protein that has been studied for more than six decades, however, its physiological role is still not completely understood. This calcium-dependent enzyme is synthesized exclusively in the liver, and found in circulation mainly associated with apoA-I in HDLs. It has been shown that PON1 association with apoA-I is necessary for optimal PON1 activity/stability [31,120].The transfer of PON1 protein from HDLs to the cell membranes of certain tissues has been hypothesized, and reported by many investigators [120,121,122,123,124,125]; but the mechanism of this transfer has not been yet ascertained. PON1 was first described in 1953 for its organophosphorus hydrolyzing properties [126,127]. Almost four decades later, a new role of PON1, in preventing the oxidation of low-density lipoproteins (LDLs) [128] and HDLs [129], was reported, generating an increased interest in the study of PON1 in relation to cardiovascular disease among the scientific community. PON1 was also found to play an important role in HDL-mediated macrophage cholesterol efflux [130]. Since then, PON1 has been studied in a myriad of other oxidative stress-related diseases and in innate immunity (reviewed in [25,131]). Although the physiological substrate of PON1 remains unknown, PON1 can hydrolyze a wide range of substrates, ranging from certain organophosphate compounds [132,133], to aromatic and cyclic carbonate esters [134], lipo-lactones (lactones are believed to be the native substrate of paraoxonases) [135,136,137], and quorum sensing factors [138,139].

PON1 belongs to a multigene family of closely related enzymes that in mammals includes paraoxonase 2 (PON2) and paraoxonase 3 (PON3) [140]. PON2 is the only member not found in circulation. All the paraoxonases are polymorphic enzymes, with more than 200 single nucleotide polymorphisms (SNPs) described in the coding and 5′ to 3′ untranslated regions of the *PON1* gene (SeattleSNPs, https://pga.gs.washington.edu/PON1). There are two polymorphisms in the coding region, *PON1_L55M_* and *PON1_Q192R_*, and one in the promoter region, *PON1_T-107C_*, which have been the focus of most studies due to their effects on PON1 concentration and/or PON1 activity [141,142,143,144]. The *PON1_Q192R_* polymorphism was the first one described, and the most extensively studied to date, due to its effect on PON1 catalytic efficiency towards certain substrates. The presence of polymorphisms on the *PON1* gene has led, and continues to lead, to many publications studying the effects of *PON1* polymorphisms on disease; almost all of them reporting conflicting results, including the few existing studies on PON1 and AD. Although some of the described polymorphisms have an effect on PON1 activity and/or concentration, there are other factors affecting PON1 levels, including age, diet, certain drugs, and lifestyle [145,146]. Therefore, the majority of these studies have disregarded the most important factor that determines susceptibility and risk to disease, which is PON1 levels and PON1 functionality. In epidemiological studies, since the two *PON1_192_* alloforms (Q and R) have quite different rates of hydrolysis of specific substrates, it is important to analyze the functional genotypes separately, using the assay named PON1 status [125,147,148].

#### 3.1.1. Major Evidence Supporting a Role of Paraoxonase 1 in Alzheimer’s Disease Pathogenesis

Oxidative stress is involved in the mechanisms of neurodegeneration, leading to AD development and progression. PON1’s antioxidant activity, its presence in both plasma and CSF [28,149,150,151], and the documented protein localization in certain regions of the brain suggest that PON1 could play a role in the pathogenesis of AD and other neurodegenerative diseases [152]. As mentioned above, most of the epidemiological studies published to date regarding PON1’s role in AD are contradictory, mostly due to the studies focusing on *PON1* SNPs analyses rather than PON1 levels (Table 4) [149,153]. This is due to *PON1* polymorphisms not being the only factor affecting PON1 activity and concentration. Studies that have focused on plasma/serum PON1 activity in relation to AD have mostly reported a consistent association between low PON1 activity in plasma or serum and AD [27,28,151,154,155,156,157]. Thus, low levels of plasma PON1 activity could be a risk factor for AD. This was not a surprising finding, as a decrease of PON1 activity with the progression of other oxidative stress-related diseases had already been reported [158,159,160,161,162,163,164]. It is likely that increased levels of oxidative stress somehow alters the structure of PON1 or the environment of PON1 in a way that decreases its enzymatic activity, resulting in a systemic decrease of antioxidant capacity and progression of disease [165,166]. In addition to plasma/serum, PON1 is also present in smaller amounts in CSF, as first reported by Wills and colleagues [150]. There is no known PON1 expression in the brain, however it is not known how PON1 is transferred from the liver to the CSF. One hypothesis is that it would cross the BBB bound to discoidal HDLs, which may enter the CNS via scavenger receptor class B, type I-mediated uptake or other unknown mechanisms [167,168]. Aside from the unknown mechanism of PON1 transfer to the CSF, there have been very few studies measuring PON1 activity in the CSF in relation to AD [151], as well as to other neurological diseases. In these studies, the levels of PON1 activity in CSF were significantly lower than those in plasma/serum, and therefore, it was difficult to assess if there was a significant decrease of CSF PON1 activity with the progression of disease [28,150]. Intriguingly, in our work we found that PON1-arylesterase/apoA-I ratio correlated with t-tau in the AD group [155]. More research into PON1 activity and PON1 protein in CSF is needed to determine if those measurements could be used as biomarkers of AD, and their diagnostic value, as well as to provide more information regarding the role of PON1 in the CSF, and in AD.

#### 3.1.2. Functional Interaction between PON1 and apoA-1: Significance in Alzheimer’s Disease

It is now well recognized that PON1 can only encounter its substrates and catalyse their transformation in vivo if the enzyme is distributed with HDL [31]. It is also becoming increasingly apparent that apoA-I and PON1 form an actual “functional couple”, with one influencing the activity of the other, and vice versa [18,31]. This consideration mostly applies to PON1, which is present in a small fraction of HDLs, while apoA-I is a constant key constituent of the lipoprotein. The widely suggested physical interaction between apoA-I and PON1 within HDL structure underlies this mutual modulation [29,166]. However, it is fair to recognize that, despite great progress in this area, our knowledge of the precise interfacial activation of the two proteins still remains limited.

Combined data from functional, crystallographic, and site-specific mutagenesis studies, has concluded that PON1 activity strictly depends on its binding to HDL phospholipids and apoA-I [169]. More specifically, data reported from Huang et al. [166], suggested that two regions on apoA-I (the so-called P1 and P2 sequences) play a direct role in PON1-HDL interaction, and are fundamental for the maintenance of PON1 function. Moreover, the catalytic site of PON1 appears to be in marked proximity to the HDL-anchoring region site. This picture, although still not completely exhaustive, is consistent with previous data. Seminal studies in this field have already highlighted the close and mutual interplay between these two proteins. It was found that during isolation PON1 co-purifies with apoA-I [170]; this result was confirmed in subsequent research showing that this accessory protein is also bound to apoJ [171]. After almost a decade, Oda and coworkers [172] demonstrated, via a site specific mutagenesis approach, that apoA-I may be important in the PON1 assembly process into HDL, and functional stability.

It has been shown that apoA-I could be oxidatively modified in atherosclerotic plaques, leading to dysfunctional HDL. In particular, this modification makes the lipoprotein unable to promote cholesterol efflux and to contrast the oxidation of LDL [24]. The major culprit of this oxidative insult is myeloperoxidase (MPO) [166,173]. Indeed, HDLs isolated from patients affected by CVD contained high levels of 3-chlorotyrosine and 3-nitrotyrosine, products of the catalytic action of MPO [18,166]. At this point, PON1 comes on stage; apoA-I integrity seems to be preserved due to the antioxidant shield afforded by PON1 [166]. An elegant study by Huang et al. [166], suggested that MPO and PON1 may interact, within HDL, during inflammation, with MPO promoting damaging oxidative modifications of both PON1 and apoA-I, and PON1 inhibiting MPO activity.

The aforementioned lines of evidence clearly suggest that in epidemiological/clinical studies dealing with PON1, apoA-I levels should also be assessed. In this regard, we have recently shown that PON-arylesterase/apoA-I ratio could be a useful additional parameter [13]. Consistent with this consideration, in a recent study we found that this ratio, but not PON1 activitie, is inversely associated with CSF total tau (t-tau) and p-tau, markers of NFT pathology and neurodegeneration, respectively [151]. Moreover, our preliminary (unpublished) data suggest that PON-arylesterase/apoA-I ratio might discriminate healthy controls from AD patients better than PON1-arylesterase activity. This parameter reflects the activity of PON1 per apoA1 (which is also a surrogate marker of HDL particle) and may represent a valid measure of the biological functionality of each, single HDL. An elevated PON-arylesterase/apoA1 ratio could indicate a more effective antioxidant protection towards apoA-I, and a minor extent of oxidation of this protein. Owing to this, it is tempting to speculate that this parameter could mark more properly the quality of HDL than solely PON1 activity. This could be one of the underlying reasons for some of the contrasting findings in epidemiological/clinical studies on PON1 [174,175,176]

## 4. Conclusions

A growing body of evidence clearly indicates an association between HDL, and its main protein components, and AD pathophysiology. In particular, beyond the deeply investigated role of apoE, the apolipoproteins apoA-I and apoJ may also influence the pathogenetic processes of AD, by acting through multiple mechanisms in the CNS, including Aβ processing, clearance, and transport across the BBB. In addition, the antioxidant, anti-inflammatory, and anti-apoptotic activity of PON1 has also been suggested to play a role in the pathogenesis of AD, although data are still limited. Notably, the physical interaction between apoA-I and PON1 may explain the reciprocal influence in AD pathogenesis of these two proteins within HDL. Although wider studies will be necessary to further elucidate this aspect, the evidence collected by this review suggests both PON1 and apoA-I, and their ratio, as novel potential biomarkers useful in monitoring AD progression.

## 5. Future Direction and Perspectives

Being a relatively novel topic, further efforts should be made to draw a definitive picture of the role of HDL in AD pathogenesis. Future preclinical and clinical studies should address two, still not exhaustively solved, questions: (1) the cause–effect relationship between HDL and AD, and (2) the mechanism by which HDL may protect from AD occurrence.
(1)As we have pointed out, the classic HDL hypothesis, increasing HDL cholesterol will decrease risk of CVD, has not provided the anticipated outcomes. Instead, the field is moving towards HDL functionality based on their protein cargo, with evidence already showing the importance of functional HDL in disease, rather than HDL-C levels. In this regard, the study of functional CNS HDL-like lipoproteins is still in its infancy and has great potential for improving our understanding on the role of the CNS lipoprotein cargo in AD prevention, and in providing therapeutic perspectives to increase HDL-like particle functionality. However, before projecting this type of pharmacological approach, the causality of dysfunctional HDL in AD development should be definitively ascertained. The available epidemiological studies, being mostly cross-sectional, have not provided sufficient proofs on this. Besides, other methodological issues should be addressed, such as the standardization of the measurement of apoE, apoJ, and PON1 activity. This problem mostly concerns the measurement of PON1 activity, due to the great heterogeneity in the selection of substrates, and in the conditions of the assays used, affecting the studies on this HDL-associated protein.(2)Given the complexity of the blood-HDL proteome and lipidome, and the limited information available, future studies focusing on these two areas will be critical for not only a better understanding of the CNS HDL-like lipoprotein functions, but also for novel therapeutic strategies. Of importance is understanding the mechanism of HDL-like lipoproteins and their components in optimizing cerebrovascular health. Besides, it is pivotal to clarify how the use of imaging modalities and imaging biomarkers can facilitate detection of cerebrovascular health, AD diagnosis, and future therapeutic development. In this regard, advances in mass spectrometry sensitivity, the -omics fields, and data management of generated large datasets would ensure the feasibility of disentangling the HDL complexity in the coming years.

## Figures and Tables

**Figure 1 antioxidants-09-01224-f001:**
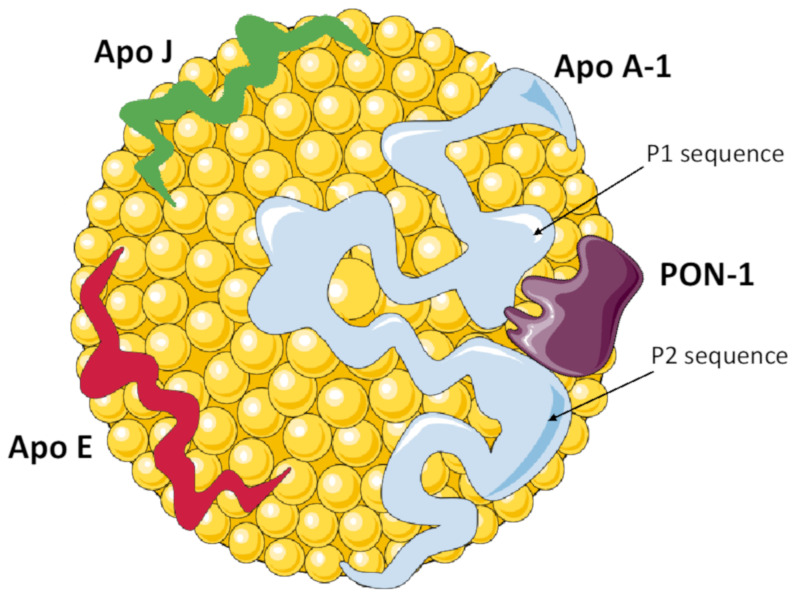
High-density lipoprotein (HDL) particle and its main proteome. ApoA-1 is the major protein constituent of HDL, playing a crucial structural and functional role. PON1 is present in a small fraction of HDLs, and its antioxidant activity strictly depends on its binding to apoA-1, through the interaction domains P1 and P2. Apo E is one of the major components of lipoprotein particles in plasma, including HDL, as well as of the HDL-like particles in the central nervous system (CNS). Apo J is expressed in different tissues and is present in biological fluids; in the CNS it is bound to HDL-like particles and seems to be involved in Aβ clearance.

**Table 4 antioxidants-09-01224-t004:** Selection of preclinical and clinical evidence supporting the role of paraoxonase 1 (PON1) in protecting from Alzheimer’s disease (AD).

Study	Type of Study	Main Findings
Clinical/Epidemiological Studies		
Cervellati, C. et al., 2015 [27]	Cross-sectional	Lower serum PON1 activity in AD, vascular dementia and MCI compared to controls (*n* = 593)
Cervellati, C. et al., 2019 [149]	Meta-analysis (genetic studies)	*PON1_S311C_* polymorphism (SS genotype) and the rs705379 (but not *PON1_T-107C_*, and *PON1_Q192R_*) were associated with risk of AD
Romani, A. et al., 2020 [151]	Cross-sectional	PON/apoA-1 ratio was inversely related to neurodegeneration (evaluated as t-tau) in AD patients; no change of CSF PON1 activity between AD and controls (*n* = 71).
Paragh, G. et al., 2002 [154]	Cross-sectional	Lower serum PON1 activity in AD and vascular dementia compared with controls (*n* = 110)
Zengi, O. et al., 2011 [155]	Cross-sectional	Lower plasma PON1 activity in AD compared to controls (*n* = 41)
Bednarz-Misa, I. et al., 2020 [156]	Cross-sectional	Serum PON1 activity correlated with the severity of AD-related cognitive decline (*n* = 237)
Bednarska-Makaruk, M.E. et al., 2013 [157]	Cross-sectional	Lower serum PON1 activity in AD, vascular dementia and mixed dementia compared to controls (*n* = 433). Higher prevalence of *PON1_T-107C_*, but not *PON1_Q192R_*, in AD patients than in controls

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
