# Peer review of "HDL Proteome and Alzheimer’s Disease: Evidence of a Link"

_antioxidants, 2020, doi:10.3390/antiox9121224_

Round 1

Reviewer 1 Report

The work is a compilation, almost exhaustive, on the important role that the levels of HDL cholesterol seems to play in the genesis and development of AD. This is obviously of interest because of the effort involved in compiling it and its value to potential readers. However, in the opinion of this reviewer, a compilation must provide some element of novelty that is extracted by the authors of the compiled set. In this case, there is no contribution whatsoever, beyond a mere transcription of the works of other authors. For example, some mechanism could have been suggested by which ApoE is anti-inflammatory or a possible competitive mechanism between HDL protein and beta-amyloid by the cholesterol molecule, could have been discussed. Furthermore, in the conclusion it is stated that (sic):  "Notably, the physical interaction between apoA-I and PON1 may explain the reciprocal influence in AD pathogenesis of these two proteins within HDL". Not even this is original since it had already been proposed by other authors (see for example Arianna Romani et al. Antioxidants (Basel). 2020 May; 9 (5): 456.).

Author Response

Comments and Suggestions for Authors

The work is a compilation, almost exhaustive, on the important role that the levels of HDL cholesterol seems to play in the genesis and development of AD. This is obviously of interest because of the effort involved in compiling it and its value to potential readers. However, in the opinion of this reviewer, a compilation must provide some element of novelty that is extracted by the authors of the compiled set. In this case, there is no contribution whatsoever, beyond a mere transcription of the works of other authors. For example, some mechanism could have been suggested by which ApoE is anti-inflammatory or a possible competitive mechanism between HDL protein and beta-amyloid by the cholesterol molecule, could have been discussed.

We acknowledge the reviewer for the useful suggestion,that we think greatly improved the content of our manuscript. The scope of our review is to summarize and critically examine the current state of knowledge on the role of selected HDL-associated proteins in AD pathogenesis and physiopathology, especially focusing on less investigated aspects and suggesting potential novel mechanisms beyond the influence on the typical disease features. Indeed, a recent review addressing together all these aspects lacks in the literature panorama. We have better specified our intent in the Introduction (lines 112-115). In addition, we have implemented the manuscript by adding additional reported mechanisms and further speculation and discussion about potential mechanisms of action supporting the involvement of HDL-proteins in Alzheimer’s disease. See lines 191-198, 277-286, 293-294, 308-311, 400-410.

Furthermore, in the conclusion it is stated that (sic): "Notably, the physical interaction between apoA-I and PON1 may explain the reciprocal influence in AD pathogenesis of these two proteins within HDL". Not even this is original since it had already been proposed by other authors (see for example Arianna Romani et al. Antioxidants (Basel). 2020 May; 9 (5): 456.)

We thank the reviewer for highlighting this point. Actually, two of the authors (Cervellati C and Zuliani G) of the present work also contributed to the study mentioned by the reviewer. We mistakenly missed to stress this in the manuscript. As now highlighted in the final paragraph of Section 3.1.2 (lines 519-524, lines 527-530) in the cited study we found a significant and inverse correlation between the PON1-arylesterase/ApoA1 ratio and CSF total tau (t-tau), and p-tau. Moreover, our preliminary (unpublished) data suggest that PON-arylesterase/apoA-I ratio might discriminate healthy controls from AD patients better than arylesterase activity (as you can see in the figure attached):

We believe that PON-arylesterase/apoA-I ratio may represent a valid and new measure of the biological functionality of each single HDL. Further studies on large samples is required to confirm the possible use of this parameter as biomarker of “HDL quality”.

Reviewer 2 Report

Thank you for the opportunity to review the manuscript. Please find below a few comments.

Very long manuscript. Please try to shorten the manuscript a bit, e.g., please try to not repeat in the text the information included in the tables.

Several reviews described the role of HDL in Alzheimer's disease. Please add the information about the novelty of the manuscript.

The citation should be added after the name of the first author, e,g. Button et al. [5].

I suggest dividing the information provided in the paragraphs "Major evidence supporting a role of Apolipoprotein A-1/E/J in Alzheimer’s disease pathogenesis" into more subparagraphs.

The tables should be prepared according to the journal guidelines.

In the manuscript, some abbreviations were explained more than one time. e.g., apo J, while other abbreviations were not explained, e.g., LRP2.

Gene names should be italicised.

"The association between apoJ concentration in brain tissue, CSF, serum and plasma, and 338 dementia has been extensively reviewed in a recent systematic review and meta-analysis [96]." Please add the information about the conclusions of the systematic review.

Please include in the manuscript directions for future studies.

Punctuation errors should be corrected:
- "..Button et al" -> please change "et al" to "et al".
- "...central nervous system (CNS." -> please add ")" after the abbreviation.

Author Response

Comments and Suggestions for Authors

Thank you for the opportunity to review the manuscript. Please find below a few comments.

Very long manuscript. Please try to shorten the manuscript a bit, e.g., please try to not repeat in the text the information included in the tables.

According to reviewer request, we have shortened all the sections of the manuscript without changing its content.

Several reviews described the role of HDL in Alzheimer's disease. Please add the information about the novelty of the manuscript.

We acknowledge the reviewer for the useful suggestion, that we think greatly improved the content of our manuscript. The scope of our review is to summarize and critically examine the current state of knowledge on the role of selected HDL-associated proteins in AD pathogenesis and physiopathology, especially focusing on less investigated aspects and suggesting potential novel mechanisms beyond the influence on the typical disease features. Indeed, a recent review addressing together all these aspects lacks in the literature panorama. We have better specified our intent in the Introduction (lines 112-115). In addition, we have implemented the manuscript by adding additional reported mechanisms and with further speculation and discussion about potential mechanisms of action supporting the involvement of HDL-proteins in Alzheimer’s disease. See lines 191-198, 277-286, 293-294, 308-311, 400-410.

The citation should be added after the name of the first author, e,g. Button et al. [5].

According to reviewers suggestion we have moved the citation after the name of the first author through all the manuscript.

I suggest dividing the information provided in the paragraphs "Major evidence supporting a role of Apolipoprotein A-1/E/J in Alzheimer’s disease pathogenesis" into more subparagraphs.

According to reviewers suggestion we divided the mentioned paragraphs in subparagraphs.

The tables should be prepared according to the journal guidelines.

The tables have been formatted according to the journal guidelines.

In the manuscript, some abbreviations were explained more than one time. e.g., apo J, while other abbreviations were not explained, e.g., LRP2.

We have accurately checked the manuscript and now the abbreviations are explained once. 

Gene names should be italicised.

Gene names have been italicized through all manuscript according to reviewer suggestion.

"The association between apoJ concentration in brain tissue, CSF, serum and plasma, and 338 dementia has been extensively reviewed in a recent systematic review and meta-analysis [96]." Please add the information about the conclusions of the systematic review.

We have added the information on conclusion of the mentioned review according to reviewer suggestion. See lines 339-345.

Please include in the manuscript directions for future studies.

We have included a section called “Future Directions and Perspectives” at the end of the manuscript, following the reviewer’s suggestion. See lines 545-571.

Punctuation errors should be corrected:

- "..Button et al" -> please change "et al" to "et al".

- "...central nervous system (CNS." -> please add ")" after the abbreviation.

The indicated punctuation errors have been corrected.

Reviewer 3 Report

Regarding the manuscript entitled “HDL proteome and Alzheimer’s disease: evidence of a link” the authors review an interesting and actual topic about Alzheimer’s disease physiopathology and pathogenesis. The review summarizes extensively the potential role of different biomarkers from HDL proteome and it association with Alzheimer’s disease.

The introduction is well structured and present the problem in a clear way.

The review is fine organized and supported by an extensive set of preclinical and clinical studies.

The conclusions are consistent with the data included in the review.

The main critic to this review is concerning the limitations of the various studies included in the review, that is very limited throughout the review. Especially at the conclusion of the article, I am of the opinion that the major limitations observed should be explored, albeit in a summarized form.

In Line 108 – a parenthesis is missing in CNS abbreviation.

Author Response

Comments and Suggestions for Authors

Regarding the manuscript entitled “HDL proteome and Alzheimer’s disease: evidence of a link” the authors review an interesting and actual topic about Alzheimer’s disease physiopathology and pathogenesis. The review summarizes extensively the potential role of different biomarkers from HDL proteome and it association with Alzheimer’s disease. The introduction is well structured and present the problem in a clear way. The review is fine organized and supported by an extensive set of preclinical and clinical studies. The conclusions are consistent with the data included in the review.

The main critic to this review is concerning the limitations of the various studies included in the review, that is very limited throughout the review. Especially at the conclusion of the article, I am of the opinion that the major limitations observed should be explored, albeit in a summarized form.

We thank the reviewer for the comment and accordingly we have added a final paragraph summarizing the major limitations of the studies examined in the review together with the future perspectives. See lines 545-571 (in particular, lines 556-562).

In Line 108 – a parenthesis is missing in CNS abbreviation.

The parenthesis has been added accordingly.

Round 2

Reviewer 1 Report

The manuscript has been significantly improved , particularly by the inclussion of the item "Future Direction and Perspectives" at the end of the work. So it is now acceptable for publication, although I continue to think that its contribution is not specially notorious. It just reinforces the well known relation between cholesterol and Alzheimer.